# Molecular Iodine Improves the Efficacy and Reduces the Side Effects of Metronomic Cyclophosphamide Treatment against Mammary Cancer Progression

**DOI:** 10.3390/ijms25168822

**Published:** 2024-08-13

**Authors:** Evangelina Delgado-González, Ericka de los Ríos-Arellano, Brenda Anguiano, Carmen Aceves

**Affiliations:** Instituto de Neurobiología, Universidad Nacional Autónoma de México (UNAM), Juriquilla 76230, Querétaro, Mexico; edelgado@comunidad.unam.mx (E.D.-G.); erios220986@comunidad.unam.mx (E.d.l.R.-A.); anguianoo@unam.mx (B.A.)

**Keywords:** iodine, metronomic chemotherapy, adjuvant, cyclophosphamide

## Abstract

Metronomic chemotherapy with cyclophosphamide (Cpp) has shown promising results in cancer protocols. These lower and prolonged doses have antiangiogenic, pro-cytotoxic, and moderate secondary effects. Molecular iodine (I_2_) reduces the viability of cancer cells and, with chemotherapeutic agents, activates the antitumoral immune response and diminishes side effects. The present work evaluates the adjuvant of oral I_2_ with Cpp using a murine model of mammary cancer. Female Sprague Dawley rats with 7,12-dimethylbenzantracene-induced tumors received Cpp intraperitoneal (50 and 70 mg/kg two times/week, iCpp50 and iCpp70) and oral (0.03%; 50 mg/Kg; oCpp50) doses. I_2_ (0.05%, 50 mg/100 mL) and oCpp50 were offered in drinking water for three weeks. iCpp70 was the most efficient antitumoral dose but generated severe body weight loss and hemorrhagic cystitis (HC). I_2_ prevented body weight loss, exhibited adjuvant actions with Cpp, decreasing tumor growth, and canceled HC mechanisms, including decreases in vascular endothelial growth factor (VEGF) and Survivin expression. oCpp50 + I_2_ diminished angiogenic signals (CD34, vessel-length, and VEGF content) and proinflammatory cytokines (interleukin-10 and tumor necrosis factor-alpha) and increased cytotoxic (lymphocytic infiltration, CD8^+^ cells, Tbet, and interferon-gamma) and antioxidant markers (nuclear erythroid factor-2 and glutathione peroxidase). I_2_ enhances the effectiveness of oCpp, making it a compelling candidate for a clinical protocol.

## 1. Introduction

Metronomic chemotherapy is characterized by frequent and low doses of chemotherapeutic agents with no prolonged drug-free breaks [1]. This treatment was initially designed for patients with advanced cancers, showing angiogenic effects, the activation of immunocytotoxic responses, the inhibition of stem cell markers, and the induction of a tumor-arrested state [2,3]. Metronomic cyclophosphamide (Cpp) is the most studied treatment due to its ease of being administered orally and its low price. However, the most prevailing side effect of this drug is hemorrhagic cystitis (HC), which occurs in up to 32% of cases, and the risk of its occurrence increases with a longer duration of treatment [4]. The co-administration of several agents to attenuate the severity of HC has been extensively studied, but acceptable prevention still needs to be determined [5,6]. Our group previously showed that the oral supplementation of molecular iodine (I_2_) improves the antitumor response of intraperitoneal Cpp in neuroblastoma xenotransplants, decreasing angiogenesis [vascular endothelial growth factor (VEGF) and vascularity], chemoresistant markers [(Survivin and B-cell lymphoma 2 (Bcl2)], and the attenuation of the HC diminish lipoperoxidation and edema in the bladder epithelium [7]. In the present work, we analyzed the oral I_2_ supplement in the antineoplastic effect of Cpp using a murine immunocompetent model of mammary cancer induced with 7,12-dimethylbenzantracene (DMBA). Two administration routes of Cpp were evaluated: intraperitoneal (iCpp) and oral (oCpp). Our results show that although both routes of Cpp showed antineoplastic effects, the oral co-administration resulted in the best combination, exerting a significant antitumoral response (tumor size and anti-angiogenesis) with the most extensive cytotoxic immune activation cluster of differentiation 8 positive (CD8^+^) and the attenuation of HC installation, which leads us to propose it as a candidate for a clinical protocol.

## 2. Results

### 2.1. I_2_ Supplementation Exhibited Adjuvant Antitumoral Action and Prevented Side Effects of Cpp Treatments

Figure 1 shows the effects of I_2_ on body weight gain and tumor growth in the intraperitoneal (i) and oral (o) administration of mCpp. Compared to the control, the supplementation of I_2_ alone did not alter body weight gain. Intraperitoneal concentrations of Cpp exhibited a moderated decrease in body weight gain at iCpp50 (10%), but a significant and severe loss in iCpp70 (24%) was observed from the first week (Figure 1A). The co-administration of I_2_ totally prevented body weight loss at the dose of 50 mg (iCpp50 + I_2_), and partially (60%) in the highest dose (iCpp70 + I_2_). In the oral treatment, the administration of Cpp50 exhibited a significant loss in body weight (18.3%), which became significant after the second week. The presence of I_2_ in continuous administration with oCpp50 exhibited a substantial preventive effect, diminishing body weight by only 9.3% (Figure 1B). 

Concerning tumor growth, the supplementation of I_2_ alone exhibited an arresting effect of 74% in comparison to the final growth in the control group (~500%) (Figure 1C,D). In the intraperitoneal doses, the tumor growth exhibited a discrete but not significant decrease in iCpp50 (10%) and a significant action in the iCpp70 group, rising to 98% inhibition. The co-administration of I_2_ exerts an adjuvant effect only in the iCpp50 + I_2_ group, exhibiting the same significance observed in both iCpp70 groups (Figure 1C). In the oral administration, the direct antineoplastic effect of oCpp is significant (92%), and the co-administration with I_2_ exhibited a discrete co-adjuvant, diminishing the tumor size by 98%.

### 2.2. I_2_ Supplement Exerts Adjuvant Apoptotic Actions and Reduces the Expression of Survival Markers in Cpp Treatments

Figure 2 shows that the I_2_ supplement did not modify the induction of apoptosis (Bax/Bcl-2 index) but significantly diminished the Survivin expression. In contrast, at the highest doses of Cpp (iCpp70 and oCpp50), a significant increase in apoptosis induction (Bax/Bcl2 index) is observed, whereas the Survivin expression has not changed at any dose. The co-administration of I_2_ in all doses of Cpp maintained the inhibition of the expression of Survivin and intensified the induction of apoptosis (Bax/Bcl2 index) in concordance with the major reduction in the tumor size (Figure 2A–D). As was expected, the VEGF expression is reduced in all treatments compared with the control group, and animals co-treated with I_2_ showed a lesser expression of this messenger (Figure 2E,F).

Peroxisome proliferator-activated receptor gamma (PPARγ), as a differentiation marker, did not exhibit changes in the protein content associated with I_2_ alone, but displayed a clear tendency to increase at iCpp70 and iCpp70 + I_2_ (Figure 2G). In the oral model, the oCpp50 group had a significant reduction in the protein expression of this receptor (Figure 2H), which was prevented in the presence of iodine (oCpp50 + I_2_).

The subsequent experiments were performed only in the groups with oral Cpp administration.

### 2.3. I_2_ and oCpp Reproduce the Metronomic Antiangiogenic Effect

To analyze the effect of oCpp50 and I_2_ administration on tumoral vascularity, the number and length of the blood vessels [(immunodetected by the cluster of differentiation 34 (CD34)], as well as the VEGF protein content, were quantified. Decreases in all parameters accompanied all conditions (Figure 3); however, the co-administration (oCpp50 + I_2_) exhibited the lowest values, suggesting the cancelation of new blood vessel formation.

### 2.4. I_2_ Supplementation Induces a Cytotoxic Environmental Response

Figure 4 shows representative micrographs and the quantification of immune responses. Figure 4A shows micrographs stained with hematoxylin-eosin (H&E, 20X); the insert shows (40X) the lymphocytes (hyperpigmented cells). Figure 4B shows micrographs showing CD8-positive immune cells (black arrows). The quantification analysis (Figure 4C,D) showed that I_2_ increased the presence of both lymphocytes and CD8^+^ cells in the interior of the tumor compared with both the control and oCpp50 groups, indicating that I_2_ induces the activation of cytotoxic immune responses. The protein content of Tbet and IFNγ (Figure 4E,F) did not change in the I_2_ groups, but in the oCpp50 group, these markers exhibited a significant decrease, which was restored by the combination of oCpp50 + I_2_.

### 2.5. I_2_ Supplements Induce Antioxidant Mechanisms and Reduce Inflammatory Response

To evaluate the antioxidant mechanism involved in I_2_ action, the expression of nuclear erythroid factor 2 (Nrf2) and its antioxidant effector, the glutathione peroxidase (GPx), were analyzed. Figure 5A shows that I_2_ administration alone or with oCpp50 did not modify the Nrf2 expression but the GPx expression increased in both groups (Figure 5B). In contrast, the control and oCpp50 groups did not exhibit changes in these parameters. The interleukin 10 (IL-10) and tumor necrosis factor-alpha (TNFα) were inhibited by the presence of I_2_, corroborating the anti-inflammatory effect of this halide (Figure 5C, D). 

### 2.6. I_2_ Supplement Exerts Direct and Preventive Anti-Inflammatory Effects on the Bladder Epithelium

Since the more frequent side effect of Cpp treatment is the installation of HC, we analyzed the bladder histopathology (H&E). Figure 6 shows that the supplementation of I_2_ alone diminished urothelial thickness compared to the control group, suggesting that the presence of mammary tumors could be associated with irritative bladder epithelium and that the supplement of I_2_ exerts a direct anti-inflammatory action. Moreover, the administration of oCpp exacerbates the thickness of the uroepithelium, and the presence of edema was evident in these tissues (black arrows), indicating the installation of HC. The group oCpp + I_2_ exhibited a significant attenuation in inflammation, suggesting a protective antioxidant effect of I_2_.

## 3. Discussion

Metronomic therapy was proposed by Hanahan et al. in 2000 [1], but its use in the clinic is limited because the suggested doses and periods are highly variable in the different protocols [3]. However, one of the most accepted consensuses includes doses ranging from one-tenth to one-third of the maximum tolerated dose and the sustained inhibition of tumor angiogenesis, which is detected as a minimal expression of VEGF and an increase in thrombospondin 1 (TSP-1) [8,9,10]. Several studies demonstrate that this angiogenic inhibition also involves processes outside the tumor, such as the inhibition of the proliferation and circulation of endothelial cells (ECs) and endothelial progenitor cells (EPCs), which leads to less differentiation in immature endothelial cells, resulting in the cancelation of the tumor neovascularization [11,12]. In addition, it has recently been accepted that metronomic success includes more actions, such as the activation of the cytotoxic immune system, the decrease in regulated T lymphocytes, and the decrease in cancer stem cells [3,13].

In the present study, we analyzed metronomic doses intraperitoneally [one-third (70 mg) and one-fourth (50 mg)] and orally (one-fourth), considering the decrease in tumor size and the low expression of VEGF as the main factors. In addition, we investigated the effect of the I_2_ supplement alone and in both Cpp conditions, using an immunocompetent rat model that led us to analyze the activation of the immune cytotoxic response. 

The results show that regardless of the route of administration (IP and oral), all Cpp doses met the metronomic criterion, and the most evident antineoplastic response was obtained with iCpp70, which generated the smallest final tumor size and low VEGF expression. However, this dose generated the worst side effects (more significant body weight loss and the installation of HC). In clear contrast, the I_2_ supplement exerted metronomic adjuvant actions, showing a decrease in tumor size, the lowest values in VEGF expression, and a reduction in Survivin expression related to chemoresistant installation [14]. These parameters are associated with a significant induction of apoptosis (Bax/Bcl2 index). In fact, the oCpp50 + I_2_ duo seems to be the best combination, since, in addition to the adjuvant antineoplastic effects, the lowest values in VEGF content were accompanied by an apparent reduction in tumor vascularity. The effect of I_2_ on the ECs and EPCs has yet to be analyzed. However, our results are consistent with several reports showing that the co-administration of metronomic schedules with antiangiogenic agents diminishes these endothelial precursors and improves the antineoplastic response, extending disease-free life [15,16]. Furthermore, in relation to the specific mechanism of I_2_ in these processes, our group has shown that this chemical form of iodine binds to arachidonic acid abundantly present in breast tumor cells and forms the iodolipid known as 6-iodolactone, which is a stimulating ligand of PPARγ receptors [17]. These receptors are closely related to suppressing cancer cell proliferation, metastasis, and chemoresistance [18].

In addition, our results also show that this oCpp50 + I_2_ exerts a clear increase in the cytotoxic response (CD8^+^ lymphocytes, Tbet, and IFNγ). In fact, one characteristic of I_2_ in the tumor microenvironment is the induction transition of the immune response from Th2 to Th1. In a previous pilot protocol on breast cancer [19], the tumoral transcriptomic analysis showed that the presence of I_2_ alone or with conventional chemotherapy induces the reactivation of antitumor immune responses, increasing the number of CD8^+^, dendritic and B-lineage cells. The mechanisms postulated for these actions could include the increase in the expression and activation of PPARγ in close coactivation of interferon regulatory factor 1 (IRF1). Initially identified as a transcriptional regulator of interferon genes in cancer, IRF1 is required for Th1 polarization in NK cells, CD8^+^ cells, and M1 macrophages. Moreover, IRF1 is associated with a decrease in Treg cell regulation, leading to the reactivation of an antitumor immune response [20]. Moreover, in chemotherapy/I_2_-treated groups, a significant upregulation in Th1 differentiation-associated genes (Tbet /TBX21, IL12RB1, IL12RB2, STAT1, STAT4, TNF, and LTA) and a substantial downregulation in genes involved in Th2 differentiation (GATA3, TSLP, and BHLHE41) were observed [19]. In accordance with these facts, in the present work, I_2_ seems to exert similar effects in terms of maintaining the expression of Tbet and IFNγ that were significantly decreased in oCpp alone. In addition to these direct immune pathways, it has been documented that I_2_ can also modify the expression of target genes such as Tbet and GATA3 through epigenetic regulation (demethylation/methylation) [21]. 

The second objective of our study was to analyze the effect of I_2_ supplements on the side effects of Cpp, which were monitored in general by weight loss and specifically by the inflammation of the bladder epithelium. Effectively, body weight loss was observed in both high doses of Cpp (iCpp70 and oCpp50), but the loss was initiated early (day 7) and was more accelerated in the iCpp70 group. In contrast, although the total dose of Cpp administered orally during the three weeks of treatment was higher than the intraperitoneal doses (230 vs. 180 mg/kg), the continuous distribution (daily for three weeks, instead of two doses every third day) exhibited less harmful effects, showing a slow and moderated body weight loss (18.3 vs. 24%). All groups of Cpp + I_2_ exhibited less body weight loss. The preventive effect of I_2_ has been documented in several of our previous studies, suggesting a general antioxidant action in chronic inflammation processes generated by doxorubicin [22] or Cpp [7]. This general effect is also evident in the prevention of bladder injuries. Cpp is a prodrug that, through hepatic biotransformation, generates the phosphoramide mustard (the antineoplastic metabolite) and the acrolein, which causes HC [4]. In the present study, the histological evaluation showed a thickening of the urothelium and the lamina propria of the bladder, suggesting inflammation and the establishment of moderate HC. The I_2_ supplement in co-administration with oCpp prevented these alterations. This protective mechanism could be due to its antioxidant effect. Acrolein that accumulates in the bladder activates xanthine oxidase and aldehyde dehydrogenase, which gives rise to the formation of free radicals, such as peroxynitrite [4]. Previously, it has been shown that the chemical form of I_2_ has, in vitro, a reducing capacity ten times greater than ascorbic acid and 60 times greater than potassium iodide (FRAP test) [22]. In vivo studies have shown that I_2_ supplementation decreases the oxidative potential in the serum of rodents and human patients [21]. The results of this work showed that this halide could be operated in several routes, directly diminishing the reactive oxygen species (ROS), activating the Nrf2 pathways, and increasing the antioxidant enzyme GPx, or through PPARγ activation, generating a low-inflammatory microenvironment in several tissues decreasing the expression of proinflammatory cytokines like IL-10 and TNF-α. This action has been described with other antioxidants, such as resveratrol [5] and carvedilol [23], reducing ROS levels in bladder tissues generated by Cpp treatment. 

Another alternative previously suggested by us [7] is that I_2_ could bind directly to acrolein, decreasing its irritating action on the bladder epithelium. This alternative is based on the structure of acrolein [21], which contains double bonds that are capable of being iodinated. Although this proposal needs to be analyzed, it is consistent with the protective mechanisms used clinically by the administration of sodium 2-mercaptomethanesulfonate (mesna). This is a sulfhydryl compound that binds to the methyl group of acrolein, leading to the formation of thioether, which is non-toxic and nullifies the irritative and inflammatory process. However, the effectiveness of the use of mesna is currently being discussed, since an extensive literature study needs to support the preventive effect of this treatment at conventional doses of Cpp [6]. Therefore, there is still a great need to search for new compounds that may be active in alleviating Cpp-induced cystitis.

According to our research, the I_2_ supplement alongside metronomic oCpp can improve its effectiveness and prevent potential inflammatory side effects. This combination can be considered a viable and cost-effective alternative to traditional hospital-based treatments, with potential for use in clinical protocols.

## 4. Materials and Methods

### 4.1. Reagents

7,12-dimethylbenzantracene (DMBA) was obtained from Sigma-Aldrich (St. Louis, MO, USA). Sublimated iodine was purchased from Macron-Avantor (Center Valley, PA, USA). Cyclophosphamide (Cpp; Hidrofosmin^®^) was obtained from Sanfer, Estado de México, México. 

### 4.2. Animals 

Fifty female Sprague–Dawley rats (130–150 gr body weight, BW; five per group) were kept under regulated temperature conditions (22 ± 1 °C) at 50% humidity on a 12 h light/12 h darkness cycle and permitted ad libitum access to food (Purina Certified Rodent Chow) and water. All the procedures followed by the Animal Care and Use Program (National Institutes of Health, Bethesda, MD, USA) were approved by the Research Ethics Committee (Protocol #31A) of the Instituto de Neurobiología at the Universidad Nacional Autónoma de México. Animals consumed 40 mL/day of drinking water, I_2,_ and Cpp solutions, and all groups were conformers of five animals.

### 4.3. Tumor Induction

The tumor induction was performed according to the Thompson method [24]. Cancer induction by the carcinogen DMBA is a widely used model for the study of epithelial mammary cancer. DMBA is metabolized by cytochrome P450 enzymes and free radical intermediates, generating DNA adducts through covalent binding to exocyclic amino groups of purines or the induction of oxidative stress. Mammary tumors are generated 8 to 15 weeks post-administration. However, tumors subsequently could appear in other regions, such as the ovary and the retina. In the present study, all rats were inoculated with a single intragastric dose of 20 mg/mL DMBA dissolved in corn oil. Eight weeks later, the animals were checked on every week to detect the presence of tumors. Once the tumors appeared and reached an approximate volume of 1 cm^3^, they were separated into different groups for their respective treatments. Body weight and tumor growth were registered every week for three weeks. Tumor volume was calculated using the ellipsoid formula [V = (major diameter x minor diameter^2^)/2]. After the period of treatment, the animals were euthanized, and the mammary tumors and bladders were collected; one part of the tumor was frozen in dry ice and stored at −80 °C for molecular analyses. The other part of the tumors and bladders were stored and fixed in buffered formalin and processed for histologic assessments.

### 4.4. Treatments

#### 4.4.1. Intraperitoneal Treatment

Two doses of Cpp were assessed: 50 and 70 mg/kg BW (iCpp50 and iCpp70, respectively), which corresponded to one-third and one-fourth of the Cpp maximum tolerated dose. Cpp was dissolved in saline solution and administered intraperitoneally two times per week for three weeks. The animals were divided into control, I_2_ (0.05%; 50 mg/100 mL in drinking water), iCpp50, iCpp50 + I_2_, iCpp70, and iCpp70 + I_2_ groups. 

#### 4.4.2. Oral Treatment

The oral doses of Cpp50 (oCpp) and I_2_ were supplemented alone or together in deionized water, and their concentrations were estimated, considering that each animal consumes around 40 mL/day. The concentration of oCpp was 0.03% (30 mg/100 mL equivalent to 50 mg/kg), and the I_2_ was 0.05% (50 mg/100 mL). The groups were control, I_2_, oCpp50, and oCpp50 + I_2_ groups. The water bottle was replaced two times a week.

### 4.5. Gene Expression

Total RNA was isolated from tumors using the Trizol reagent method (Invitrogen, Waltham, MA, USA). cDNA was synthesized with 2 ug of RNA using the MML-V system (Thermo Fisher, Waltham, MA, USA). Real-time PCR was carried out using Maxima Sybr green mix (Thermo Fisher) with the Rotor-Gene 3000 thermocycler from Corbett Research (Concord, New South Wales, Australia). Relative expression was normalized to the mRNA level of β-actin. The information on the oligonucleotides used is summarized in Table 1.

### 4.6. Protein Expression

Frozen tumors were homogenized in a RIPA buffer, and the total protein was quantified using the Bradford method. These extracts were used for Western blot and multiplex analysis.

#### 4.6.1. Western Blot

Protein homogenates from the tumors (50 ug) were separated by electrophoresis in 10% [for the detection of peroxisome proliferator-activated receptor (PPARγ), nuclear erythroid factor 2 (Nrf2), or T-box transcription factor 21 (Tbet)] or 20% [for glutathione peroxidase (GPx) or interferon-gamma (IFNγ) detection] acrylamide gel. The proteins were later transferred to nitrocellulose membranes. The unspecified reaction was blocked for 2 h with 0.1% Tween 20-Phosphate-buffered saline (PBS-T) containing 5% skimmed milk powder. The membranes were incubated with their corresponding antibodies diluted in PBS-T with 0.1% skimmed milk powder overnight at 4 °C. Antibody information is shown in Table 2. The secondary antibodies were incubated for 2 h in PBS-T with 0.1% skimmed milk powder. The proteins were visualized with the Clarity chemiluminescent system (Bio-Rad, Hercules, CA, USA); the blots were scanned for densitometric analysis, which was performed using Image J V1.53e software (National Institutes of Health, Bethesda, MD, USA). The protein levels were normalized to the total protein (Ponceau staining). 

#### 4.6.2. Multiplex Analysis 

The multiplex quantification of tumor cytokines interleukin-10 (IL-10), tumor necrosis factor-alpha (TNF-α), and vascular endothelial growth factor (VEGF) was performed using the ProcartaPlex Mix & Match assay kit (Lot 349558-000, Thermo Fisher, Waltham, MA, USA), according to the manufacturer’s protocol. Briefly, protein extracts from the tumors (10 mg/mL) and the corresponding standards were incubated overnight at 4 °C with the immunobead mix. A biotinylated detection antibody was added and incubated for 30 min at room temperature. Afterward, streptavidin-phycoerythrin was added for posterior reading in the xMAP instrument Luminex 200 (Luminex Corporation, Austin, TX, USA). The data were collected from the instrument for calculation and analysis using the Procartaplex analysis application.

### 4.7. Histologic Evaluation

Histopathologic and immunohistochemical images were visualized under a light microscope (Leica DM250, Wetzlar, Germany), and pictures were acquired using a digital camera (Leica DFC 420); the analysis was performed using Image J V1.53e software.

#### 4.7.1. Hematoxylin-Eosin Staining

Paraffin sections of the tumors and bladders were stained with hematoxylin-eosin (H&E) for histopathological analysis. Lymphocyte infiltration in the tumors and urothelial thickness were measured in three random fields.

#### 4.7.2. Immunohistochemistry

Specific antibodies (Table 2) were used for endothelial cells (CD34) and active lymphocytes (CD8^+^). Sections were counterstained with hematoxylin. The signal was detected using the ABC and DAB systems (Vector Laboratories, Newark, CA, USA). Positive cells and blood vessel length were computed in three random fields.

### 4.8. Statistical Analysis

The results were expressed as means ± SD. We used one-way ANOVA and Tukey’s post hoc tests to determine the significant differences between the groups (*p* < 0.05). *, ** or different letters indicate statistical differences between groups. Statistical analyses were performed with GraphPad Prism v6.01 (GraphPad Software, La Jolla, San Diego, CA, USA). 

## 5. Conclusions

Our study shows that I_2_ supplementation in cyclophosphamide metronomic therapies decreases chemoresistance mechanisms and increases the apoptotic response, and significantly attenuates chemotherapy side effects. The possibility of offering an oral, outpatient metronomic treatment with few side effects is a desirable proposition both for the patient’s quality of life and for the hospital costs associated with breast cancer treatment. This combination is an attractive proposal for clinical studies.

## Figures and Tables

**Figure 1 ijms-25-08822-f001:**
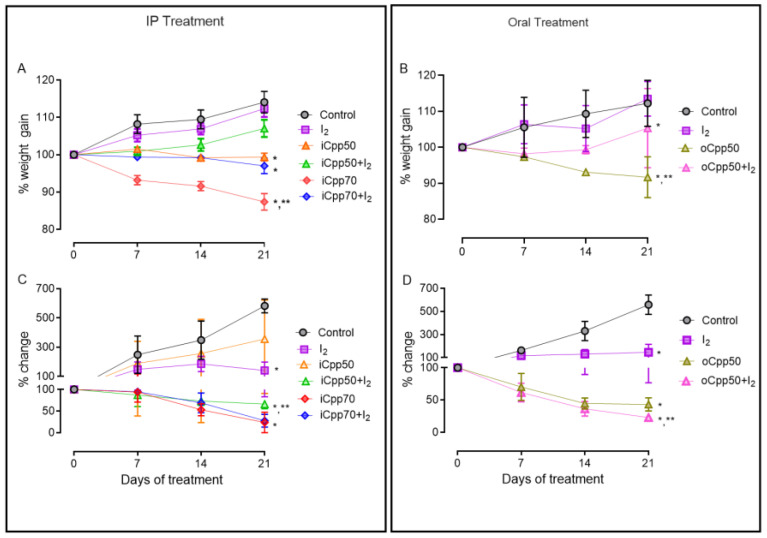
Body weight and tumor growth in rats with mammary cancer. The graphs show the percentage of change in body weight gain in response to intraperitoneal (**A**) or oral (**B**) treatments. (**C**,**D**) show the percentage of tumor growth. Tumor volume was calculated by the ellipsoid formula: Volume = (major diameter x minor diameter^2^)/2. iCpp, intraperitoneal metronomic chemotherapy; oCpp, oral metronomic chemotherapy. Points represent means ± SD *, statistical differences (*p* < 0.05) when compared to the control group. **, Statistical differences between iCpp70 and oCpp50 and iCpp70 + I_2_ and oCpp50 + I_2_. *n* = 5/group.

**Figure 2 ijms-25-08822-f002:**
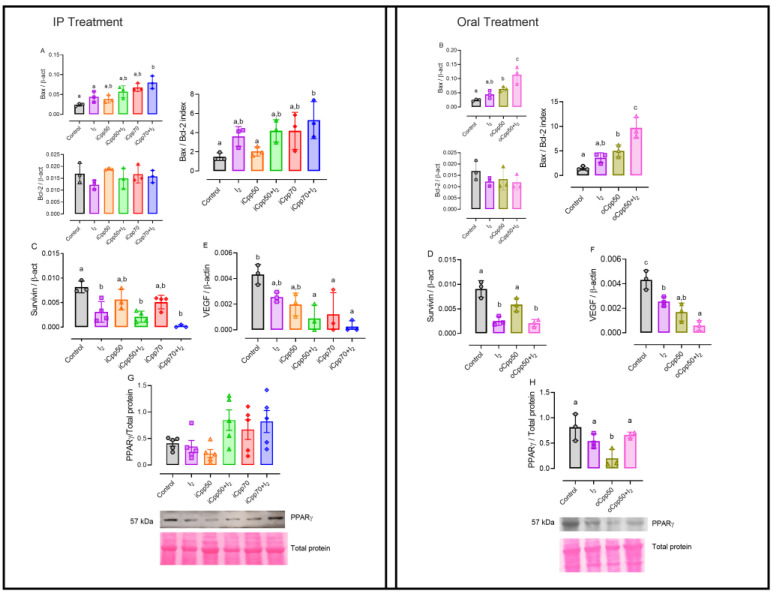
Molecular responses after 3 weeks of treatment. Expression (quantitative PCR) in tumoral samples from intraperitoneal (IP) or oral treatment. Apoptosis (**A**,**B**), chemoresistance (**C**,**D**), and angiogenesis (**E**,**F**). Immunodetection of peroxisome proliferator-activated receptor gamma (PPARγ) was analyzed by Western blotting. Expression was normalized by Ponceau staining and analyzed by band densitometry (**G**,**H**). Bax, bcl-2-like protein 4; Bcl-2, B-cell lymphoma 2; VEGF, vascular endothelial growth factor. Graphs representing mean ± SD. Data were analyzed using variance analysis (ANOVA) and Tukey post hoc test. Different letters indicate statistical differences (*p* < 0.05). n = 3–5/group.

**Figure 3 ijms-25-08822-f003:**
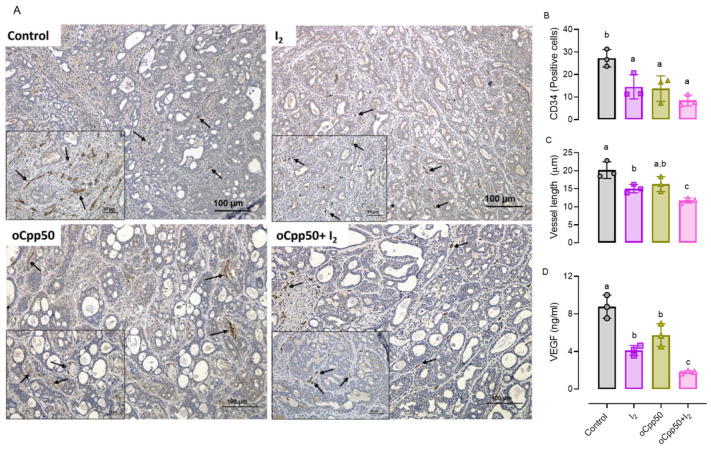
Vascularity and angiogenesis in tumoral samples of oral treatment. (**A**): micrographs (20X) stained with anti-clusters of differentiation 34 (CD34; positive cells, black arrows). The insert shows (40X) the lymphocytes (hyperpigmented cells). (**B,C**): quantification of positive cells and vessel length (um), each value representing the average of three different fields: (**D**): multiplex protein content of VEGF. Graphs represent means ± SD. ANOVA and Tukey post hoc test were used to analyze data. Different letters indicate statistical differences (*p* < 0.05). n = 3/group.

**Figure 4 ijms-25-08822-f004:**
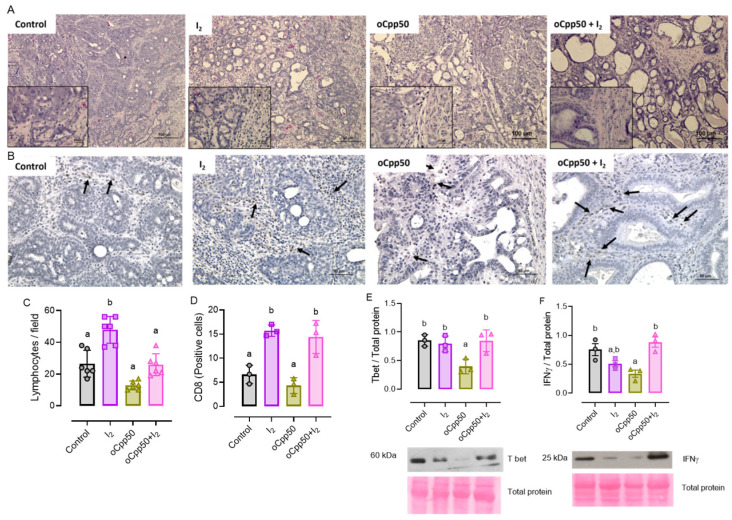
Antitumoral immune response to oral metronomic chemotherapy (oCCp). (**A**): hematoxylin-eosin (H&E) staining of tumors to identify lymphocyte infiltration (small round hyperpigmented cells). Micrographs (20X) and magnification (40X) are shown. (**B**): micrographs of CD8 positive cells (black arrows; 20X). (**C**): quantification of lymphocytic infiltration by average of three random fields. (**D**): number of positive cells to CD8, computing average of three random fields (20X). (**E**,**F**): immunodetection of Tbox transcription factor 21 (Tbet) and interferon-gamma (IFNγ) by Western blot. Expression was normalized by Ponceau staining and analyzed by band densitometry. Graphs represent means ± SD. ANOVA and Tukey post hoc tests analyzed data. Different letters indicate statistical differences (*p* < 0.05). n = 3/group.

**Figure 5 ijms-25-08822-f005:**
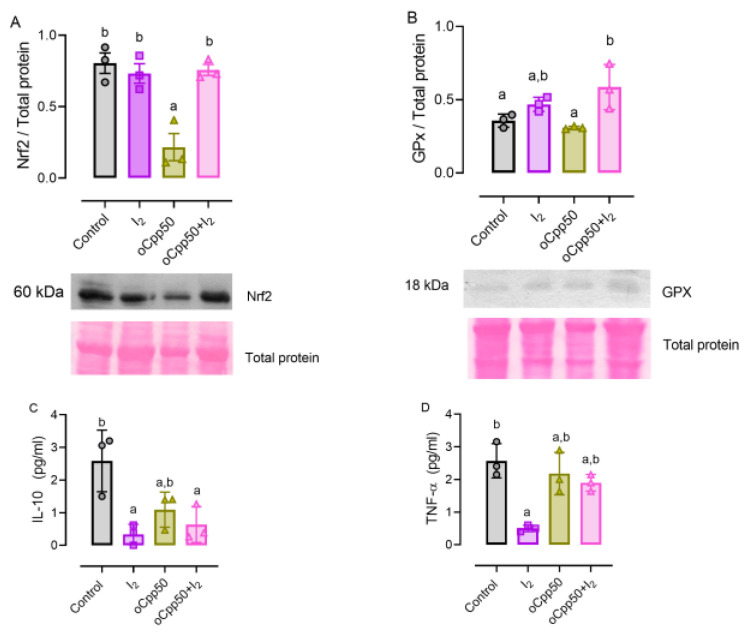
Antioxidant and anti-inflammatory response to oCpp in rat mammary tumors. (**A**): nuclear erythroid factor 2 (Nrf2); (**B**): glutathione peroxidase (GPx) protein expression, as indicators of antioxidant mechanism, were evaluated by Western blotting. Expression was normalized by Ponceau staining and analyzed by band densitometry. (**C**,**D**), interleukin-10 (IL-10), and tumor necrosis factor (TNF-α) were assessed by multiplex assay. Graphs represent means ± SD. Data were analyzed using ANOVA and Tukey post hoc tests. Different letters indicate statistical differences (*p* < 0.05). n = 3/group.

**Figure 6 ijms-25-08822-f006:**
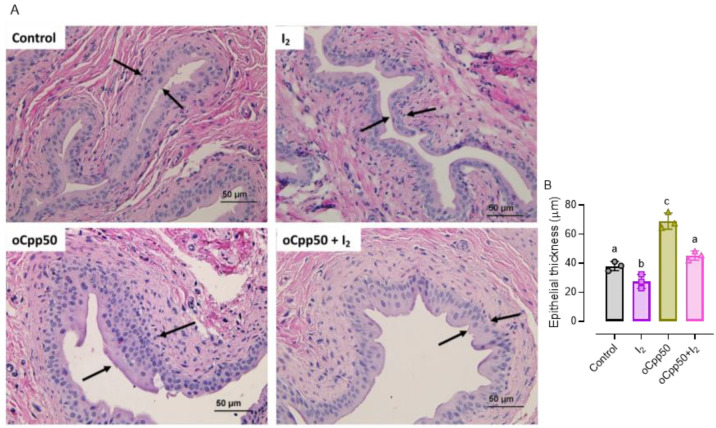
I_2_ supplementation prevents bladder inflammation associated with oCpp. (**A**): micrographs stained with H&E of bladder epithelium (40X). Black arrows signalized the epithelial thickness, which was associated with inflammation: (**B**): quantitation of the thickness of the urothelium. The graph represents means ± SD. Data were analyzed using ANOVA and Tukey post hoc tests. Different letters indicate statistical differences (*p* < 0.05). n = 3/group.

**Table 1 ijms-25-08822-t001:** Oligonucleotides used in RT-PCR.

Gene	GenBank ID	Sequence (5′ to 3′)	Size (pb)
Bax	NM_017059.2	FWD: CAGGGAGGATGGCTGGGGAGA	351
		REV: TCCAGACAAGCAGCCGCTCACG	
Bcl-2	NM_016993.2	FWD: GAGGCTGGGATGCCTTTGT	125
		REV: TGCACCCAGAGTGATGCAG	
Surv	NM_022274.2	FWD: AAGCCACTTGTCCCAGCTT	198
		REV: CTCATCCACTCCCTTCCTC	
VEGF	NM_001287113.1	FWD: TCACCAAAGCCAGCACATAG	120
		REV: TTTCTCCGCTCTGAACAAGG	
β-actin	NM_031144.3	FWD: CCATCATGAAGTGTGACGTTG	195
		REV: ACAGAGTACTTGCGCTCAGGA	

FWD: forward; REV: reverse; Bax: Bcl2-like protein 4; Bcl-2: B-cell lymphoma 2; Surv: Survivin; VEGF: vascular endothelial growth factor.

**Table 2 ijms-25-08822-t002:** Antibodies used for Western blot and immunohistochemistry.

Antibody	Code (Vendor)	Dilution
Rabbit polyclonal anti- PPARγ	ab209350 (Abcam, Cambridge, MA, USA)	1:2000
Mouse monoclonal anti-Tbet	ab91109 (Abcam, Cambridge, MA, USA)	1:1000
Mouse monoclonal anti-IFNγ	sc390800 (Santa Cruz Biotechnology, Dallas, TX, USA)	1:500
Mouse monoclonal anti-Nrf2	sc-365949 (Santa Cruz Biotechnology, Dallas, TX, USA)	1:250
Mouse monoclonal anti-GPx	ST1000 (Calbiochem, San Diego, CA, USA)	1:200
Mouse monoclonal anti-CD8	sc-7970 (Santa Cruz Biotechnology, Dallas, TX, USA)	1:100
Rabbit polyclonal anti-CD34	ab182981 (Abcam, Cambridge, MA, USA)	1:500
Goat anti-Rabbit-HRP	65-6120 (Invitrogen, Waltham, MA, USA)	1:5000
Goat anti-Mouse-HRP	62-6520 (Invitrogen, Waltham, MA, USA)	1:3000
Horse anti-mouse IgG biotinylated	BA-1000 (Vector Laboratories, Newark, CA, USA)	1:1000
Goat anti-rabbit IgG biotinylated	BA-2000 (Vector Laboratories, Newark, CA, USA)	1:1000

PPARγ; peroxisome proliferator-activated receptor gamma, Tbet; T-box transcription factor 21, IFNγ; interferon-gamma, Nrf2; nuclear erythroid factor 2, GPx; glutathione peroxidase, CD8; cluster of differentiation 8, CD34; cluster of differentiation 34, HRP; Horseradish peroxidase, IgG; Immunoglobulin.

## Data Availability

The data presented in this study are available upon specific request from the corresponding author.

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
