# Peer review of "Molecular Iodine Improves the Efficacy and Reduces the Side Effects of Metronomic Cyclophosphamide Treatment against Mammary Cancer Progression"

_ijms, 2024, doi:10.3390/ijms25168822_

Round 1

Reviewer 1 Report

Comments and Suggestions for Authors

This article investigated the therapeutic effects of molecular iodine (I2) combined with cyclophosphamide (Cpp) on a Female Sprague Dawley rats breast cancer model. This study applied various Cpp doses to treat breast cancer model rats, along with supplementation with I2. The results of the study showed that supplementation with I2 prevented weight loss, and reduced angiogenic signaling. Cpp50+I2 showed the most balanced results in terms of efficacy and mitigation of side effects, making it a promising candidate for clinical regimens. I would recommend the paper for publication in this journal with some minor revisions.

1. It is recommended that the legend in Figure 2 be enlarged for clearer reading.
2. How was the dose of I2 administered in the article determined?
3. Why were the concentrations Cpp50 and Cpp70 chosen for Cpp? Were other concentrations tried in the preexperiment?
4. The potential molecular mechanisms about how I2enhances the therapy efficiency of Cpp should be analyzed.
5. A three-week treatment period was mentioned, but no details were given as to why this treatment period was chosen or whether it adequately represented the timeframe for treatment of metastatic breast cancer in rats.
6. Please add the significance of the control group in the article.
7. It is suggested that some additional information be added to the conclusion to make it more complete.

Comments on the Quality of English Language

 Minor editing of English language required

Author Response

Please see the attachment."

Reviewer 2 Report

Comments and Suggestions for Authors

The authors analysed the co-administration of cyclophosphamide and I2 in the treatment of chemically induced breast cancer in an animal model

The work is very interesting and well conducted. My main perplexity lies in the number of animals per group. In the materials and methods section it is not specified how many animals there are per group, in the results authors generically say '3-5', but where there are 3 it seems very few. I also wonder how it is possible for there to be statistical significance in a group with 3 animals. I invite the authors to be more precise on this aspect

other items below

line 127_fig1a please amend

a general remark: it is really difficult to understand the figures of the western blots, which are very small, the total proteins are coloured with red ponceau and sometimes (especially fig. 2 and 5b) the bands are difficult to identify (and in any case very different from each other). I suggest that better quality figures be shown

I have a methodological question. If the compounds are dissolved in water, how can you be sure that the dose is fully taken up? And even if it is, in what time frame is the drug taken up? Shouldn't the change in the drug's plasma concentration due to irregular and protracted intake be taken into account?

Author Response

Please see the attachment."
